# Handling Data Gaps in Reported Field Measurements of Short Rotation Forestry

**Diana-Maria Seserman ***[ID] **and Dirk Freese**

Chair of Soil Protection and Recultivation, Institute of Environmental Sciences, Brandenburg University of Technology Cottbus–Senftenberg, Konrad-Wachsmann-Allee 6, 03046 Cottbus, Germany; freese@b-tu.de
* Correspondence: seserman@b-tu.de

**Abstract:** Filling missing data in forest research is paramount for the analysis of primary data, forest statistics, land use strategies, as well as for the calibration/validation of forest growth models. Consequently, our main objective was to investigate several methods of filling missing data under a reduced sample size. From a complete dataset containing yearly first-rotation tree growth measurements over a period of eight years, we gradually retrieved two and then four years of measurements, hence operating on 72% and 43% of the original data. Secondly, 15 statistical models, five forest growth functions, and one biophysical, process-oriented, tree growth model were employed for filling these data gap representations accounting for 72% and 43% of the available data. Several models belonging to (i) regression analysis, (ii) statistical imputation, (iii) forest growth functions, and (iv) tree growth models were applied in order to retrieve information about the trees from existing yearly measurements. Subsequently, the findings of this study could lead to finding a handy tool for both researchers and practitioners dealing with incomplete datasets. Moreover, we underline the paramount demand for far-sighted, long-term research projects for the expansion and maintenance of a short rotation forestry (SRF) repository.

**Keywords:** missingness; data gap; statistics; amelia; yield-safe

---

## 1. Background and Summary

Biomass generated from dedicated energy crops, such as short rotation forestry (SRF), is gaining recognition as a flexible source of energy, heat, fuel, bio-based materials, and chemicals [1,2]. SRF refers to growing fast-growing tree species, planted at a high density [3], and harvesting the trees in rotations of two to six years in order to produce woody biomass [4]. Hence, the planting popularity of fast-growing tree species belonging to the genera *Populus*, *Eucalyptus*, *Pinus*, *Acacia*, and *Salix* has increased as a result of the progressively higher demand for woody biomass for energy purposes [5–11].

In order to support such demands, management decisions in the practice of SRF require systematic measurements of trees for repository and database monitoring. Paramount for forest managers are accurate estimates of tree height, root height diameter (RHD), and breast height diameter (BHD) for the determination of timber volume [12].

However, forest research is oftentimes confronted with missing data in field measurements [12] due to sampling infeasibilities (bad weather, lack of equipment, lack of technical expertise), sampling irregularities across years, inaccurate estimations (allometric functions and the vast amount of methods that can be used for calculating one parameter), or failure of equipment (dendrometers, lysimeters, station maintenance) [13]. Moreover, the available studies performed on SRF describe annual data values, collected over a few years from tens or hundreds of tree measurements, occasionally together with the standard deviation and number of samples, which nevertheless, leads to condensed annual information about the actual growth characteristics.

Missing data not only represent a loss of information and a source of uncertainty in data analysis, but also a severe drawback for present investigations, as well as for future decision-making, coping management strategies, risk assessments, and adaptation scenarios. Therefore, filling missing data in forest research is paramount for the analysis of primary data, forest statistics, land use strategies, as well as for the calibration/validation of forest growth models.

Heretofore, little to no investigation has been carried for handling missing forestry data. In a study performed by Diamantopolou [12], tree diameters were inferred for one year with the help of a database containing more than 440 measurement points and under several artificial neural network models. However, studies on handling missing forestry data under a reduced sample size (e.g., yearly values collected over five to ten growing years) and for management practices such as SRF are rare.

Consequently, our main objective was to investigate several methods for filling missing data under a reduced sample size. Firstly, we employed a complete dataset containing yearly first-rotation tree growth measurements in an SRF area over a period of eight years, as reported by Bärwolf et al. [14]. From this complete dataset, we gradually retrieved two and then four years of measurements, hence operating on 72% and 43% of the original data. Secondly, 15 statistical models, five forest growth functions, and one biophysical, process-oriented, tree growth model were employed for filling these data gap representations accounting for 72%, and 43% of the available data.

Differences between the investigated models were addressed in order to identify the most appropriate way to retrieve information about the trees from existing yearly measurements. Subsequently, the findings of this study could lead to finding a handy tool for both researchers and practitioners dealing with incomplete datasets. Moreover, we underline the paramount demand for far-sighted, long-term research projects for the expansion and maintenance of an SRF repository.

## 2. Data Description

As a case study, we employed reported measurements of first-rotation hybrid poplar trees (*Populus nigra* L. × *P. maximowicii* A. Henry, clone Max) established near Dornburg, Germany (N51°01′N, E11°39′; 260 m a.s.l.), on around 2 ha of a total area of 51.3 ha, and managed as short rotation forestry (SRF) [14]. The poplar trees were planted in March 2007 at a planting density of 2,200 trees per hectare (i.e., at a tree spacing of 1.5 m × 3 m). With the exception of 2007, yearly measurements of height (H), root height diameter (RHD, measured at the height of 0.1 m above the ground), and breast height diameter (BHD, measured at the height of 1.3 m above the ground) were collected by the Thuringian Center for Renewable Resources, Thuringian State Institute for Agriculture from the end of vegetation period 2008 to 2014.

Therefore, the initially reported data consisted of seven points collected from the end of vegetation period 2008 (winter season 2009) to the end of vegetation period 2014 (winter season 2015). The available range of observed data (i.e., n = 869, 975, 1350, and 1357 measurement values collected at the beginning of 2009, 2010, 2011, and 2012, respectively) is represented by the standard deviation. From these original data (100%), we randomly retrieved data accounting for two years (28%) and four years (57%) in order to create representations that simulate data gaps (Figure 1).

In addition to separating the original data into two data gap representations containing 72% and 43% of the available data, we also separated the analysis between the progression of an individual parameter over time and the progression of an individual parameter depending on another parameter. This way, the analysis discerned between data missing completely at random (MCAR, i.e., values that are randomly missing from an original dataset do not relate to each other, and there is no pattern to the actual values of the missing data) and data missing at random (MAR, i.e., values that are randomly missing from an original dataset and relate to other variables two by two).

Due to the physical and physiological factors that exist between tree dimensions in forest stands [15], the relationship between variables was considered nonlinear throughout the study.

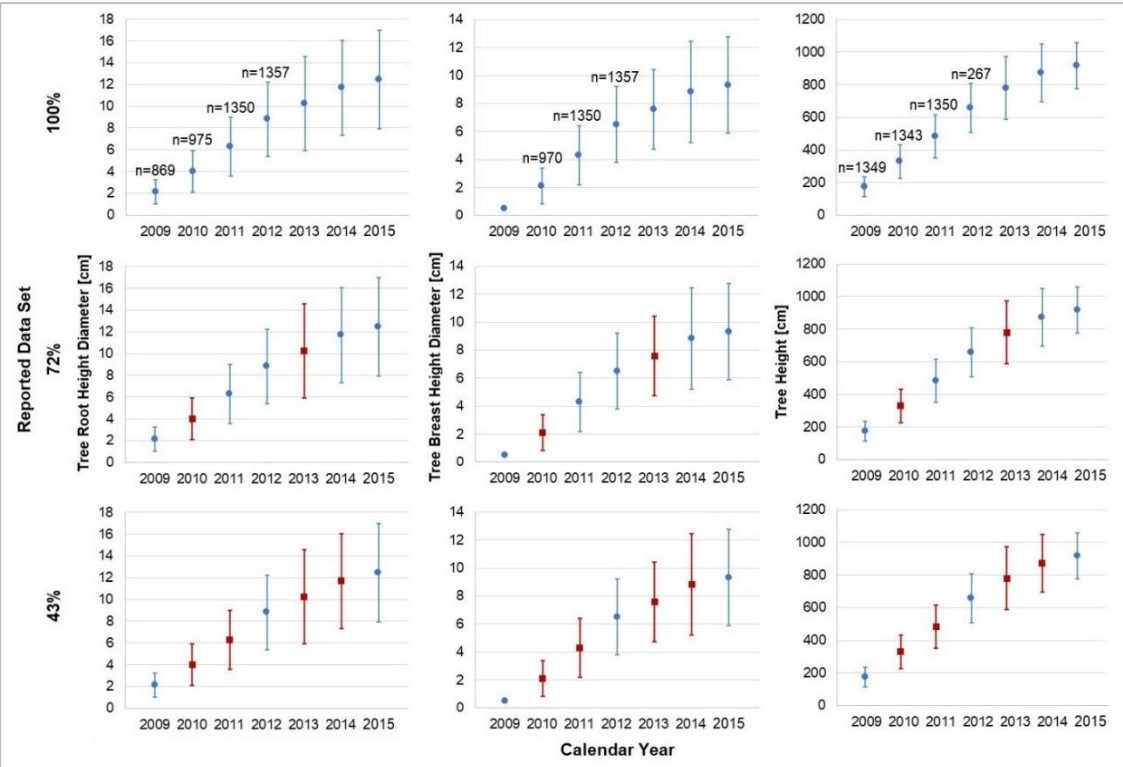

**Figure 1.** The complete dataset (100%), together with two data gap representations containing 72% and 43% of the original data for the tree root height diameter, breast height diameter, and tree height. The data are represented by the average value (blue circles and red squares for the existing and missing data, respectively), as well as by the standard deviation (blue and red error bars for the existing and missing data, respectively) and the sample size (n), when available.

## 3. Methods

For the objective of this study, we neglected the listwise deletion, an approach where missing observations are removed, and focused on imputing those missing values from the existing part of the data. In statistics, imputation is an approach where missing data are substituted, thus creating a standard method for handling missing data [16]. Nevertheless, we did not use zero or constant imputation, which replace the missing value with either zero or a constant value, respectively.

However, when filling missing data only through statistical imputations, the temporal resolution remains dependent on the existing data. Therefore, we also investigated the possibility of retrieving information on a finer temporal resolution (monthly, daily) from existing yearly measurements, with the help of a biophysical, process-oriented, tree growth model. Collectively, our analysis employed (i) regression analysis, (ii) statistical imputation, (iii) forest growth functions, and (iv) a tree growth model, which accounted for the competition for resources between trees.

### 3.1. Regression Analysis

Regression analysis is a part of inference statistics where relationships between parameters are examined. Thus, a "best fit" function (curve) with minimum residuals is assigned to the existing data points. For the conciseness of this study, standard textbooks for inference biostatistics methods by Linder [17], Mudra [18], and Rasch [19] are recommended.

Accordingly, ten regression models (Table 1) were applied to the established data gap representations by using the "Curve Fitting Toolbox" from MATLAB (version R2017a, Mathworks).

**Table 1.** The investigated regression analysis models.

| Model Name | General Model |
|---|---|
| Exponential | $a \times \exp(b \times x)$ |
| Fourier | $a_0 + a_1 \times \cos(x \times w) + b_1 \times \sin(x \times w)$ |
| Gaussian | $a_1 \times \exp(-((x - b_1)/c_1)\text{\textasciicircum}2)$ |
| Power: one term | $a \times x\text{\textasciicircum}b$ |
| Power: two terms | $a \times x\text{\textasciicircum}b + c$ |
| Rational | $(p_1)/(x + q_1)$ |
| Sum of Sine | $a_1 \times \sin(b_1 \times x + c_1)$ |
| Linear Fit | $a \times (\sin(x - pi)) + b \times ((x - 10)\text{\textasciicircum}2) + c$ |
| Polynomial: first degree | $p_1 \times x + p_2$ |
| Polynomial: second degree | $p_1 \times x\text{\textasciicircum}2 + p_2 \times x + p_3$ |

In this section, only MCAR data were taken into consideration because the data size of the response variable had to be the same as that of the predictor variable. Accordingly, the progression of an individual parameter was investigated over time, together with the progression of an individual parameter depending on another parameter.

*3.2. Interpolation*

Interpolation is a part of inference statistics where an exact fit to the existing data points is identified. Accordingly, four regression models (Table 2) were applied to the established data gap representations by using the "Curve Fitting Toolbox" from MATLAB (version R2017a, Mathworks).

**Table 2.** The investigated interpolation models.

| Model Name | General Model |
|---|---|
| Interpolant: Nearest Neighbor | |
| Interpolant: Linear | Piecewise polynomial |
| Interpolant: Cubic | computed from p. |
| Interpolant: PCHIP (Piecewise Cubic Hermite Interpolation) | |

As in the previous section, only MCAR data were taken into consideration, and the progression of an individual parameter was investigated over time, as well as its progression depending on the other parameters.

*3.3. Multiple Imputation*

Substituting missing values by multiple imputations can represent a general-purpose approach for handling missing data. Just as the name suggests, this approach creates multiple substitutes for a missing data point from all the information present in an existing dataset [20].

One of the most robust and accessible multiple imputation programs is represented by the Amelia II R-Package [20]. By employing Amelia II and the R software (version 3.4.2, R Core Team 2017), both MCAR data and MAR data were taken into consideration in this section for analyzing both the progression of an individual parameter over time and the progression of an individual parameter depending on another parameter.

*3.4. Forest Growth Functions*

Some of the essential forest growth functions [15] are presented in Table 3. Other functions were not added to this list because they resemble a general model presented previously, in Section 3.1. For example, the diameter-height relationship proposed by Assmann [21] ($H = a_0 + a_1 \times d + a_2 \times d\text{\textasciicircum}2$) resembles the second-degree Polynomial, and the allometric function [15] ($a \times x\text{\textasciicircum}b$) resembles a

Power function with one term. For this section, we used the MATLAB environment (version R2017a, Mathworks).

**Table 3.** The investigated forest growth functions.

| Model Name | General Model |
| --- | --- |
| Assmann [21] | $H = a + b \times \ln D$ |
| Korsun [22] | $H = \exp(a_0 + a_1 \times \ln(D) + a_2 \times \ln(D)^2)$ |
| Michailoff [23] | $H = a_0 \times \exp(-a_1/D) + 1.3$ |
| Petterson [24] | $H = (D/(a_0 + a_1 \times D))^3 + 1.3$ |
| Prodan [25] | $H = D^2/(a_0 + a_1 \times D + a_2 \times D^2) + 1.3$ |

H: height; D: diameter.

In this case, only MAR data were used and only for analyzing the progression of tree height with respect to tree diameter.

### 3.5. Process-Oriented Tree Growth Model

In a thorough review, Pretzsch et al. [26] compared 54 forest growth models in terms of characteristics and interactions that occur in forests both at an individual tree level and stand level. Accordingly, several models were identified as able to simulate the growth of poplar trees with respect to the specific tree phenology, light, and water availability. However, since this study focused on retrieving information about tree growth from a limited availability of data, the Yield-SAFE model was employed.

The Yield-SAFE model (Yield Estimator for Long-term Design of Silvoarable AgroForestry in Europe) is a parameter-sparse, biophysical and eco-physiological, process-oriented model, developed for growth processes in forestry, agriculture, and agroforestry systems [27–29]. Heretofore, the Yield-SAFE model has been calibrated and validated for poplar, walnut, cherry, holm oak, and stone pine trees in the Atlantic and Mediterranean regions of Europe [29–33].

The main reasoning behind choosing the Yield-SAFE model, as implemented in MATLAB, stood in its ability to render robust results under a scarcity of data [28]. In order to calibrate the model, a set of parameters and inputs was required, namely weather data (daily averages of temperature, precipitation, and global radiation) over the investigated growth period, and site-specific soil and tree parameters.

The weather data were gathered from the DWD (Deutscher Wetterdienst) station Weimar (station ID: 05419). The tree parameters were set according to annual reports [14] and adapted from literature [28,29]. The clay and sand contents were 28%, and 8%, respectively [14], which classified the soil texture as "medium-fine" [34], and hence the Mualem-van Genuchten soil parameters were set accordingly from existing estimations [34]. Collectively, the tree and soil parameters used in the Yield-SAFE model are shown in Table A1 (Appendix A). In this section, only the progression of the RHD over time was substituted as part of the MAR data.

### 3.6. Statistical Analysis

The performance of the investigated models was evaluated by the coefficient of determination ($R^2$), sum of squared errors (SSE), root-mean-square error (RMSE), mean absolute error (MAE), as well as by the concordance correlation coefficient (CCC) and simulation bias (SB) from the observations. In order to account for the variability of observations and to avoid overfitting, the best fit of the investigated models was chosen with respect to the closeness to the average, as a representative for the location of the majority of measurements.

A fit was considered useful for prediction when $R^2$ values were closer to 1.0 and when SSE, RMSE, and MAE values were closer to 0.0. Regarding the CCC and SB, a study performed by Ojeda et al. [35] proposed the following labels for model validation: "very good" for CCC > 0.90 and

SB < 20%, "satisfactory" for 0.75 < CCC < 0.90 and 20% < SB < 30%, "acceptable" for 0.60 < CCC < 0.75 and 30% < SB < 40%, and "poor" for the rest of the cases. Together with these recommendations, the performance of the models was categorized in terms of the CCC and SB, as well as in terms of the $R^2$, SSE, RMSE, and MAE, while striving for a normal distribution of residuals.

## 4. Results and Discussion

Regarding the complete reported dataset, the individual data points for each year were spread out over a wide range of values, especially for the diameter measurements (Figure 1), amounting to around ±25% for the tree heights and around ±40% for both tree diameters, as compared to their respective average values. This is, however, the case for many fast-growing tree species such as poplar (*Populus* spp.) [36,37] and black locust (*Robinia Pseudoacacia* L.) [36], which show great growth variability, even when planted at the same time and on the same land area. By comparison, first-rotation poplar trees established at the experimental site Neißetal reported a standard deviation of about ±30% for the tree heights and about ±40% for the root height diameter (RHD) over three years of growth [36]. At Wendhausen, the measurement variability of breast height diameter (BHD) amounted to around 39%, as compared to the average of all values and over six growing years [37].

### 4.1. Regression Analysis

Most of the investigated ten regression models were able to fit a curve to the existing data within the limits set by the standard deviation, except for Exponential and one-term Power models (due to an inability to fit the progression of tree dimensions over time) and the Fourier model (due to an inability to fit when subjected to the 43% data gap representation). For the conciseness of the paper, the goodness of validation of all applied regression models is presented in Table A2.

Generally, the Gaussian model performed the best, being closely followed by the Power model with two terms, Sum of Sine, and the first-degree Polynomial, then the second-degree Polynomial. The lower end was represented by the Rational, Fourier, Exponential, and one-term Power models in descending order, and finally the Linear Fitting.

As an example of the capability of the models to infer missing data from the available 43% data gap representation of RHD, three regression models labelled as "very good" (Gaussian), "satisfactory" (Sum of Sine), and "poor" (Linear Fitting) are selected and presented in Figure 2.

Visible differences can be noticed between the three regression models presented in Figure 2, justifying the requirement of several regression evaluation metrics and highlighting the importance of even slight differences between them. While the $R^2$, SB, and CCC suggested a "very good" performance of all three models, they were labelled as "very good" (Gaussian), "satisfactory" (Sum of Sine), and "poor" (Linear Fitting) given the differences in SSE, RMSE, and MAE. Even if the deviations from the fitted curve to the observations seem small, significant biases can later arise from further calculations.

### 4.2. Interpolation

All of the investigated interpolation models were capable of finding a fit encompassed in the range of observations. However, when striving for the average value, as a central tendency for most of the observations, the interpolation models generally performed the best to worse in the following sequence: Linear > Cubic > PCHIP > Nearest Neighbor. For the conciseness of the paper, the goodness of validation of all applied interpolation models is shown in Table A3.

However, since the SSE, RMSE, and MAE values were rather high when fitting the tree height over time, the 72% data gap representation of tree height was furtherly examined (Figure 3).

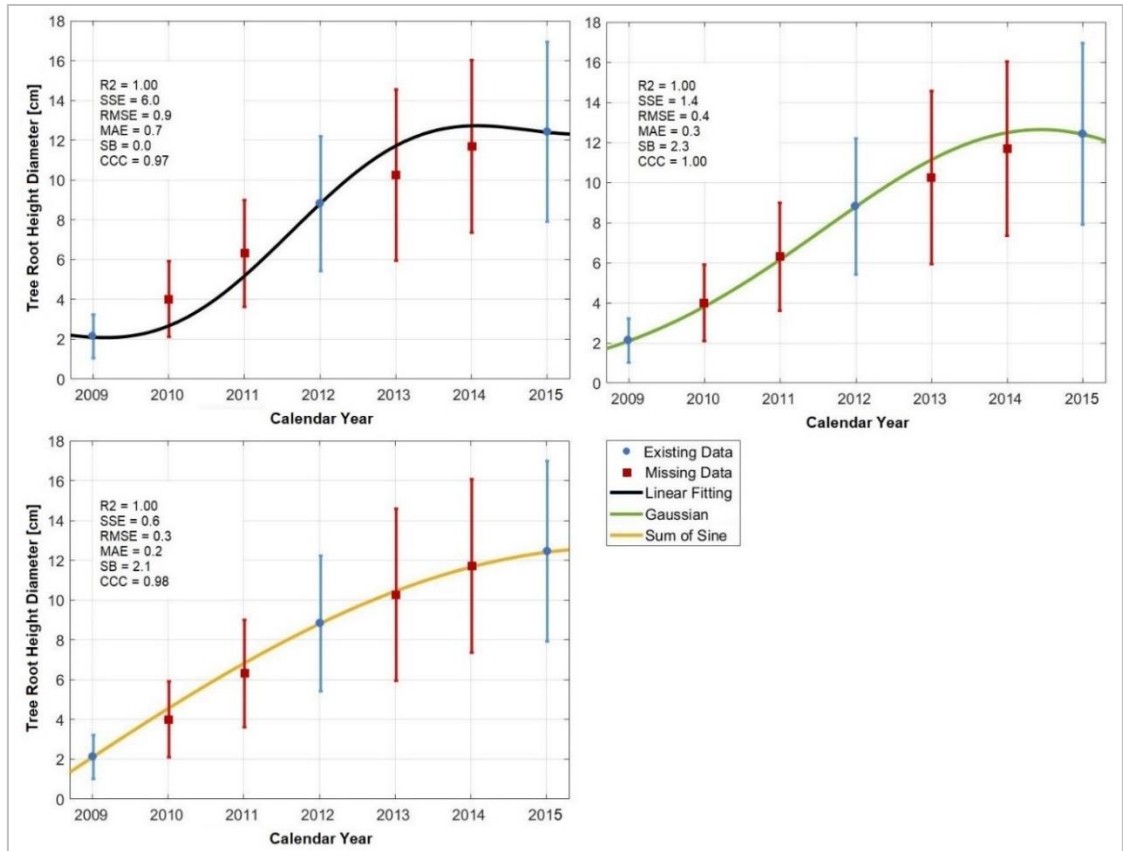

**Figure 2.** The Linear fitting (black line), Gaussian (green line), and Sum of Sine (orange line) regression models applied to the tree root height diameter (RHD) data gap representation accounting for 43% of the original dataset, in terms of the coefficient of determination ($R^2$), sum of squared errors (SSE), root-mean-square error (RMSE), mean absolute error (MAE), as well as the concordance correlation coefficient (CCC) and simulation bias (SB) from the observations. The error bars (blue and red for the existing and missing data, respectively) represent the standard deviation of the RHD over the investigated period.

Since interpolation techniques generally find an exact fit to the existing data, they assume no measurement errors, suggesting that their applicability to real-life scenarios is limited. Moreover, while interpolation is easy and fast, it does not consider the correlations between features, and it does not account for the uncertainty in the imputations.

Therefore, even if Figure 3 displays rather accurate estimations of tree height over time and under limited data availability, the four interpolation models were labelled as performing "very good" (Linear, Cubic, PCHIP) and "poor" (Nearest Neighbor) when applied to the 72% tree height data gap representation.

### 4.3. Multiple Imputation

Most of the imputations rendered by Amelia II fell within the whole range of measurements. However, when considering the majority of values, only a few results of Amelia II were labelled as performing better than "acceptable" (Table 4).

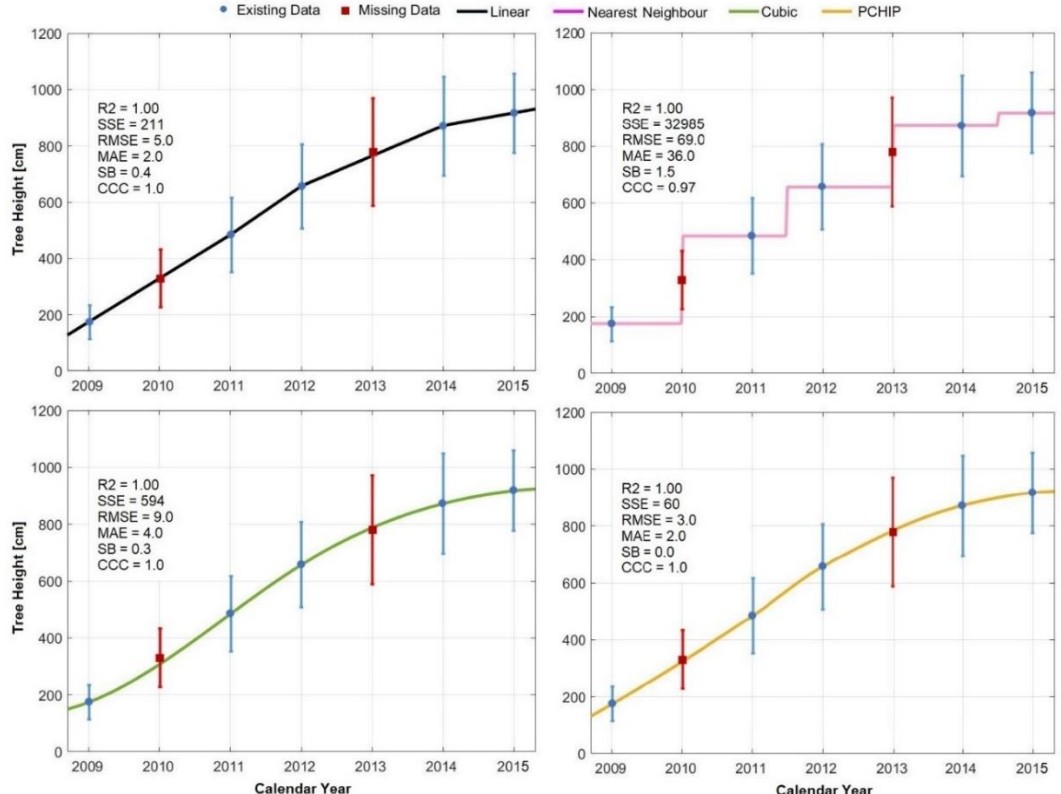

**Figure 3.** The Linear (black line), Nearest Neighbor (pink line), Cubic (green line), and PCHIP (orange line) interpolation models applied to the 72% tree height data gap representation together with the coefficient of determination ($R^2$), sum of squared errors (SSE), root-mean-square error (RMSE), mean absolute error (MAE), as well as the concordance correlation coefficient (CCC) and simulation bias (SB) from the observations. The error bars (blue and red for the existing and missing data, respectively) represent the standard deviation of the tree height over the investigated period.

**Table 4.** The goodness of validation of Amelia II in terms of the coefficient of determination ($R^2$), sum of squared errors (SSE), root-mean-square error (RMSE), mean absolute error (MAE), as well as the concordance correlation coefficient (CCC) and simulation bias (SB) from the observations.

| Model | Variable | Data Gap Representation | $R^2$ | SSE | RMSE | MAE | SB [%] | CCC | Label |
|---|---|---|---|---|---|---|---|---|---|
| Amelia II | RHD | 72 | 1.00 | 0.2 | 0.3 | 0.3 | 6.4 | 1.00 | Satisfactory |
| | | 43 | 1.00 | 4.1 | 1.0 | 0.8 | 14.9 | 0.00 | Poor |
| | BHD | 72 | 0.99 | 0.7 | 0.6 | 0.6 | 11.5 | 0.97 | Poor |
| | | 43 | 0.99 | 0.4 | 0.3 | 0.3 | 5.5 | 0.99 | Acceptable |
| | Height | 72 | 0.99 | 10,110.2 | 71.1 | 57.3 | 14.1 | 0.94 | Poor |
| | | 43 | 0.99 | 24,478.6 | 78.2 | 70.4 | 4.3 | 0.92 | Satisfactory |
| | BHD & RHD | 72 | 1.00 | 0.9 | 0.7 | 0.6 | 15.1 | 0.97 | Poor |
| | | 43 | 0.99 | 5.0 | 1.1 | 1.0 | 3.6 | 0.44 | Poor |
| | Height & BHD | 72 | 1.00 | 1627.5 | 20.5 | 20.4 | 10.1 | 0.98 | Poor |
| | | 43 | 0.98 | 2986.8 | 19.6 | 16.7 | 7.6 | 0.98 | Poor |
| | Height & RHD | 72 | 1.00 | 1214.8 | 17.7 | 17.3 | 3.6 | 0.99 | Poor |
| | | 43 | 0.99 | 3065.4 | 19.8 | 17.9 | 4.7 | 0.49 | Poor |

RHD: root height diameter; BHD: breast height diameter.

### 4.4. Forest Growth Functions

Most of the investigated forest growth functions were labelled as delivering a "very good" performance (Table 5). Accounting for all regression evaluation metrics, the Korsun [22] model fit the best, followed by Michailoff [23] and Petterson [24], Prodan [25], and then Assmann [21].

**Table 5.** The goodness of validation of forest growth functions in terms of the coefficient of determination ($R^2$), sum of squared errors (SSE), root-mean-square error (RMSE), mean absolute error (MAE), as well as the concordance correlation coefficient (CCC) and simulation bias (SB) from the observations.

| Model | Variable | Data Gap Representation | $R^2$ | SSE | RMSE | MAE | SB [%] | CCC | Label |
|---|---|---|---|---|---|---|---|---|---|
| Assmann [21] | Height | 72 | 0.99 | 3.68 | 0.73 | 0.64 | 3.8 | 0.97 | Poor |
| | BHD | 43 | 0.98 | 4.77 | 0.83 | 0.66 | 7.5 | 0.96 | Poor |
| | Height | 72 | 0.99 | 5.49 | 0.89 | 0.79 | 1.8 | 0.75 | Poor |
| | RHD | 43 | 0.99 | 7.10 | 1.01 | 0.84 | 6.1 | 0.70 | Poor |
| Prodan [25] | Height | 72 | 0.98 | 1.51 | 0.46 | 0.34 | 4.3 | 0.99 | Satisfactory |
| | BHD | 43 | 0.98 | 91.19 | 3.61 | 2.22 | 25.1 | 0.68 | Poor |
| | Height | 72 | 1.00 | 0.10 | 0.12 | 0.10 | −0.3 | 1.00 | Very good |
| | RHD | 43 | 1.00 | 0.13 | 0.14 | 0.10 | 0.2 | 1.00 | Very good |
| Petterson [24] | Height | 72 | 0.97 | 1.70 | 0.49 | 0.33 | 4.2 | 0.99 | Acceptable |
| | BHD | 43 | 0.97 | 1.65 | 0.49 | 0.34 | 4.0 | 0.99 | Acceptable |
| | Height | 72 | 1.00 | 0.10 | 0.12 | 0.08 | −0.5 | 1.00 | Very good |
| | RHD | 43 | 1.00 | 0.10 | 0.12 | 0.09 | 0.3 | 1.00 | Very good |
| Korsun [22] | Height | 72 | 1.00 | 0.11 | 0.13 | 0.08 | 1.1 | 1.00 | Very good |
| | BHD | 43 | 1.00 | 0.11 | 0.13 | 0.07 | 1.0 | 1.00 | Very good |
| | Height | 72 | 1.00 | 0.12 | 0.13 | 0.11 | −0.2 | 1.00 | Very good |
| | RHD | 43 | 1.00 | 0.11 | 0.13 | 0.09 | 1.1 | 1.00 | Very good |
| Michailoff [23] | Height | 72 | 0.99 | 0.92 | 0.36 | 0.23 | 3.5 | 0.99 | Satisfactory |
| | BHD | 43 | 0.98 | 0.90 | 0.36 | 0.25 | 3.2 | 0.99 | Satisfactory |
| | Height | 72 | 0.99 | 0.62 | 0.30 | 0.25 | −1.7 | 1.00 | Very good |
| | RHD | 43 | 0.99 | 0.76 | 0.33 | 0.27 | −1.2 | 1.00 | Satisfactory |

RHD: root height diameter; BHD: breast height diameter.

## 4.5. Process-Oriented Growth Model

If not for a high simulation bias from observations in the first years (Figure 4), the performance of the Yield-SAFE model would have been generally labelled as "very good" for this experimental site (Table 6).

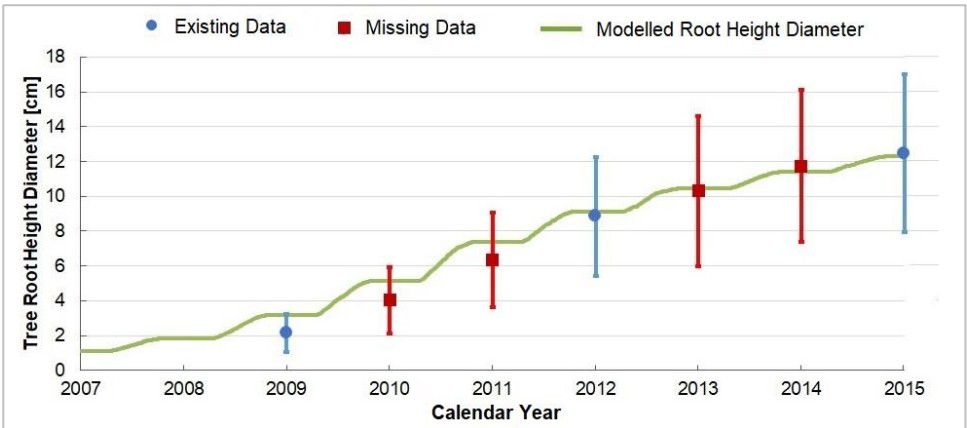

**Figure 4.** The tree root height diameter, as simulated by the Yield-SAFE model (green line) given the existing (blue circles) and missing data (red circles). The error bars (blue and red for the existing and missing data, respectively) represent the standard deviation of the root height diameter over the investigated period.

The Yield-SAFE model rendered "satisfactory" correspondences with the measured tree root height diameters under 43% availability of data, with deviations between 17% and 51% in the first three years and between 1% and 4% in the following four years.

Notable to this section was that, by using a biophysical, process-oriented model, there was a possibility of retrieving information on a finer temporal resolution (monthly, daily) from existing yearly measurements. Moreover, information about other parameters became available, such as the woody biomass and the soil water content of the site throughout the growing period (Figure 5).

**Table 6.** The goodness of validation of Yield-SAFE in terms of the coefficient of determination ($R^2$), sum of squared errors (SSE), root-mean-square error (RMSE), mean absolute error (MAE), as well as the concordance correlation coefficient (CCC) and simulation bias (SB) from the observations.

| Model | Variable | Data Gap Representation | $R^2$ | SSE | RMSE | MAE | SB [%] | CCC | Label |
|-------|----------|------------------------|-------|-----|------|-----|--------|-----|-------|
| Yield-SAFE | RHD | 72 | 1.00 | 3.7 | 1.0 | 0.9 | 12.4 | 0.99 | Satisfactory |
|  |  | 43 | 1.00 | 4.0 | 1.1 | 1.1 | 15.1 | 0.99 | Satisfactory |

RHD: root height diameter.

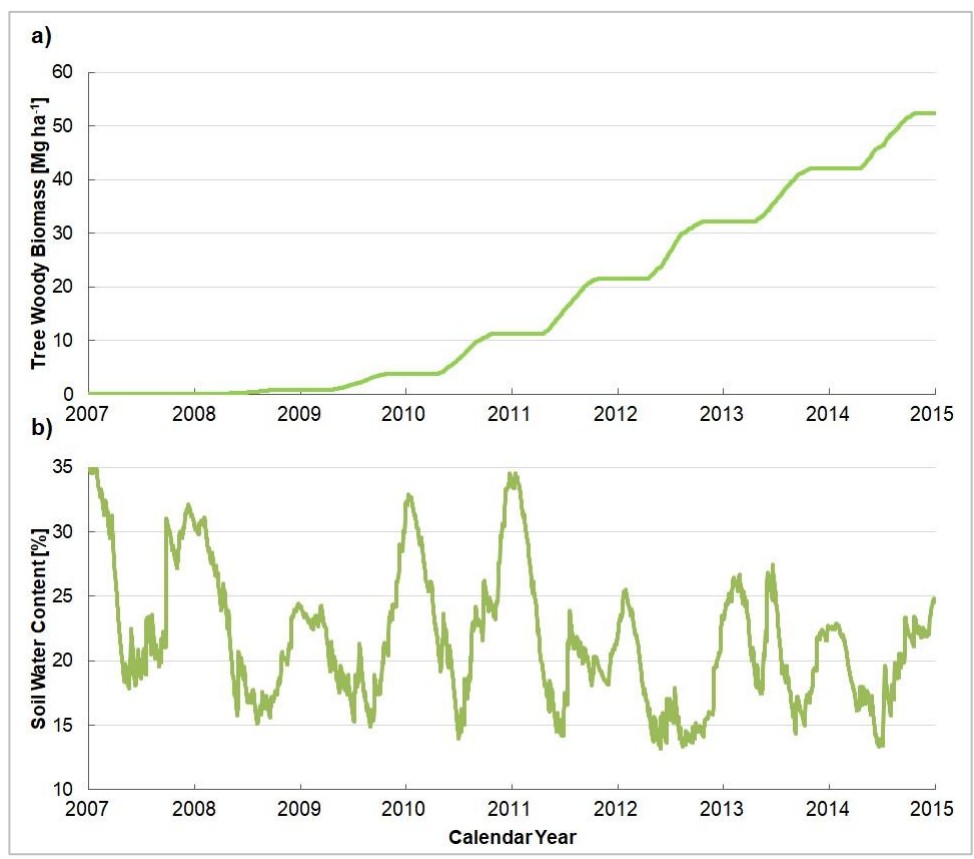

**Figure 5.** The tree woody biomass (**a**) and soil water content (**b**), as simulated by the Yield-SAFE model (green lines) from the day of planting (2007) to the day of harvest (2015).

The tree woody biomass and the soil water content, as simulated by the Yield-SAFE model, were broadly corroborated by on-site assessments. According to reported values, an average tree woody biomass of around 52 Mg ha$^{-1}$ was harvested from the poplar SRF area at the end of vegetation period 2014 (winter season 2015) [14]. Regarding the soil water content, between 13% and 24% was reported at the beginning of June 2012, and between 15% and 35% at the end of November 2012 [14].

It is notable that, while tree growth models are widely used for prediction purposes, either for future risk assessments or under different climatic, edaphic, and management scenarios, this study emphasized another role of such models, namely, for imputing gaps in knowledge.

## 5. Conclusions

This paper presented and analyzed the performance of several models belonging to (i) regression analysis, (ii) statistical imputation, (iii) forest growth functions, and (iv) a tree growth model to retrieve information about trees from existing yearly measurements. When taking into consideration the entire range of measured data, the performance of all investigated models was deemed as "very good".

However, when focusing on the area comprising most of the observations, or the central tendency of the data, as shown by the average, significant differences arose between the models.

From the curve-fitting models, the Gaussian model performed the best, being tightly followed by the Power model with two terms, Sum of Sine, and the first-degree Polynomial, then the second-degree Polynomial. Nearing this performance, the Linear, Cubic, and PCHIP interpolation models also showed good correspondences with the measurements, both under 72% and 43% data gap representations. The forest growth functions rendered good results, following the sequence: Korsun [22] > Michailoff [23] and Petterson [24] > Prodan [25] > Assmann [21]. Unsurprisingly, most of these models performed better under higher data availability, i.e., under 72% of existing data, as compared to 43% of existing data.

The Yield-SAFE model simulated the daily growth of the hybrid-poplar clone "Max I" in terms of root height diameter with satisfactory accuracy, responding sensitively to changes in the edaphic and climatic conditions. Additionally, the performance of the model was sustained by other parameters, such as the tree woody biomass and soil water content, which matched reported values. Last but not least, this study showed that a process-oriented model such as Yield-SAFE can provide descriptions of tree growth and soil water content on a finer, daily temporal scale from a scarce availability of data.

Therefore, the findings of this study could subsequently lead to finding a handy tool for both researchers and practitioners dealing with incomplete datasets. In the future, for a better understanding and reproducibility of studies, box plots should be increasingly used, showing minimums, maximums, medians, means, outliers, and the interquartile range. Moreover, we underline the paramount demand for far-sighted, long-term research projects for the expansion and maintenance of an SRF repository.

**Author Contributions:** The SIGNAL project was initiated and planned in relation to the proposal call "Soil as a Sustainable Resource for the Bioeconomy—BonaRes" (http://www.signal.uni-goettingen.de). D.-M.S. analyzed the data and wrote the manuscript. D.F. contributed by revising the manuscript.

**Funding:** This research was carried out under the BonaRes—SIGNAL project (FKZ 031A562E, 2015-2018) and funded by BMBF—German Federal Ministry of Education and Research (Bundesministerium für Bildung und Forschung).

**Acknowledgments:** We are grateful to the editor and reviewers for their time, assistance, and suggestions that helped improve this paper.

**Conflicts of Interest:** The authors declare no conflict of interest. The founding sponsors had no role in the design of the study; in the collection, analyses, or interpretation of data; in the writing of the manuscript; and in the decision to publish the results.

## Abbreviations

| | |
|---|---|
| BHD | breast height diameter |
| H | height |
| MAR | missing at random |
| MCAR | missing completely at random |
| RHD | root height diameter |
| SRF | short rotation forestry |

## Appendix A

**Table A1.** The tree and soil parameter values used for the parametrization of the Yield-SAFE model for the SRF area in Dornburg (Thuringia, Germany).

| Symbol | Description | Unit | Value | Source |
|---|---|---|---|---|
| $n_{Shoots0}$ | Initial number of shoots per tree | $tree^{-1}$ | 1.0 | Own data |
| $B_{t0}$ | Initial tree biomass | $g\ tree^{-1}$ | 40 | [38] |
| $LA_{t0}$ | Initial tree leaf area | $m^2\ tree^{-1}$ | 0.0 | [28,29] |
| $\varepsilon_t$ | Radiation use efficiency | $g\ MJ^{-1}$ | 1.04 | Own data |
| $K_t$ | Light extinction coefficient | – | 0.5 | Own data |
| $t_t$ | The number of days after budburst at which the leaf area has reached 63.2% of its maximum leaf area $LA_{ss}^{max}$ | d | 10 | [28,29] |
| $LA_{ss}^{max}$ | Maximum leaf area for a single shoot | $m^2$ | 0.05 | [28,29] |
| $n_{Shoots}^{max}$ | Maximum number of shoots per tree | $tree^{-1}$ | 10,000 | [28,29] |
| $K_{main}$ | Relative attrition rate of tree biomass | $d^{-1}$ | $10^{-4}$ | [28,29] |
| $\gamma_t$ | Transpiration coefficient of the trees | $m^3\ g^{-1}$ | 0.0002 | [38] |
| $(pF_{crit})_t$ | Critical pF value for trees | log (cm) | 4.0 | [29] |
| $(pF_{pwp})_t$ | pF value at permanent wilting point | log (cm) | 4.2 | [29] |
| $DOY_{budburst},$ $DOY_{leaffall}$ | Day of year for budburst and leaffall | DOY | 105, 300 | [38] |
| $\rho_t$ | Planting density | $trees\ ha^{-1}$ | 2200 | [14] |
| $\theta_0$ | Initial volumetric water content | $m^3\ m^{-3}$ | 0.35 | [34] |
| $\delta_{eva}$ | Potential evaporation per unit energy | $mm\ MJ^{-1}$ | 0.15 | [29] |
| $D$ | Depth of the soil compartment | mm | 1000 | [34] |
| $\alpha$ | Van Genuchten parameter | – | 0.0083 | [34] |
| $n_{soil}$ | Van Genuchten parameter | – | 1.2539 | [34] |
| $\delta$ | Parameter affecting the drainage rate below the root zone | – | 0.07 | [34] |
| $PWP$ | Permanent wilting point | log (cm) | 4.2 | [28,29] |
| $(pF_{crit})_E$ | Critical pF value for evaporation | log (cm) | 2.3 | [28,29] |
| $pF_{FC}$ | Water tension at field capacity | log (cm) | 2.3 | [28,29] |
| $K_s$ | Soil hydraulic conductivity at saturation | $mm\ d^{-1}$ | 2.272 | [34] |
| $\theta_s$ | Saturated volumetric water content | $m^3\ m^{-3}$ | 0.43 | [34] |
| $\theta_r$ | Residual volumetric water content | $m^3\ m^{-3}$ | 0.01 | [34] |

**Table A2.** The goodness of validation of all applied regression models in terms of the coefficient of determination ($R^2$), sum of squared errors (SSE), root-mean-square error (RMSE), mean absolute error (MAE), as well as the concordance correlation coefficient (CCC) and simulation bias (SB) from the observations.

| Model | Variable | Data Gap Representation | $R^2$ | SSE | RMSE | MAE | SB [%] | CCC | Label |
|---|---|---|---|---|---|---|---|---|---|
| Exponential | RHD | 72 | 0.90 | 10.3 | 1.2 | 1.1 | 1.0 | 0.93 | Acceptable |
| | | 43 | −3.31 * | 534.3 | 8.7 | 7.9 | na | na | Na |
| | BHD | 72 | −3.32 * | 287.1 | 6.4 | 5.6 | na | na | Na |
| | | 43 | −2.19 * | 287.1 | 6.4 | 5.6 | na | na | Na |
| | Height | 72 | 0.90 | 51,868 | 86 | 78 | 0.7 | 0.94 | Satisfactory |
| | | 43 | −3.58 * | 3,013,195 | 656 | 602 | na | na | Na |
| | BHD & RHD | 72 | 0.97 | 2.4 | 0.6 | 0.5 | 1.3 | 0.99 | Very good |
| | | 43 | 0.97 | 2.8 | 0.6 | 0.5 | 2.7 | 0.98 | Satisfactory |
| | Height & BHD | 72 | 0.93 | 4.0 | 0.8 | 0.6 | 1.9 | 0.97 | Poor |
| | | 43 | 0.92 | 4.1 | 0.8 | 0.6 | 1.0 | 0.95 | Satisfactory |
| | Height & RHD | 72 | 0.96 | 3.0 | 0.7 | 0.5 | 0.3 | 0.99 | Very good |
| | | 43 | 0.96 | 3.0 | 0.7 | 0.5 | 1.4 | 0.99 | Satisfactory |
| Fourier | RHD | 72 | 1.00 | 0.3 | 0.2 | 0.2 | 0.3 | 0.99 | Very good |
| | | 43 | na | na | na | na | na | na | Na |
| | BHD | 72 | 1.00 | 0.2 | 0.2 | 0.1 | 1.5 | 0.99 | Satisfactory |
| | | 43 | na | na | na | na | na | na | Na |
| | Height | 72 | 1.00 | 185 | 5.0 | 5.0 | 0.0 | 1.00 | Very good |
| | | 43 | na | na | na | na | na | na | Na |
| | BHD & RHD | 72 | 1.00 | 0.1 | 0.1 | 0.1 | 1.4 | 1.00 | Very good |
| | | 43 | na | na | na | na | na | na | Na |
| | Height & BHD | 72 | 1.00 | 0.2 | 0.1 | 0.1 | 1.4 | 0.99 | Satisfactory |
| | | 43 | na | na | na | na | na | na | Na |
| | Height & RHD | 72 | 1.00 | 0.1 | 0.1 | 0.1 | 0.2 | 1.00 | Very good |
| | | 43 | na | na | na | na | na | na | Na |
| Gauss | RHD | 72 | 0.99 | 0.4 | 0.2 | 0.2 | 0.1 | 0.99 | Very good |
| | | 43 | 1.00 | 1.4 | 0.4 | 0.3 | 2.3 | 1.00 | Very good |
| | BHD | 72 | 0.98 | 1.2 | 0.4 | 0.4 | 2.9 | 0.99 | Very good |
| | | 43 | 1.00 | 7.1 | 1.0 | 0.7 | 5.5 | 0.96 | Acceptable |
| | Height | 72 | 1.00 | 1627 | 15 | 15 | 0.3 | 1.00 | Very good |
| | | 43 | 1.00 | 4578 | 26 | 18 | 1.1 | 1.00 | Very good |
| | BHD & RHD | 72 | 1.00 | 0.4 | 0.2 | 0.2 | 1.4 | 1.00 | Very good |
| | | 43 | 1.00 | 0.6 | 0.3 | 0.2 | 1.7 | 0.99 | Satisfactory |
| | Height & BHD | 72 | 0.99 | 0.8 | 0.3 | 0.3 | 2.0 | 0.99 | Very good |
| | | 43 | 1.00 | 1.9 | 0.5 | 0.4 | 0.6 | 0.98 | Acceptable |
| | Height & RHD | 72 | 1.00 | 0.3 | 0.2 | 0.2 | 0.2 | 1.00 | Very good |
| | | 43 | 1.00 | 0.4 | 0.3 | 0.2 | 0.1 | 1.00 | Very good |

**Table A2.** *Cont.*

| Model | Variable | Data Gap Representation | $R^2$ | SSE | RMSE | MAE | SB [%] | CCC | Label |
|---|---|---|---|---|---|---|---|---|---|
| Power: one term | RHD | 72 | na | na | na | na | na | na | Na |
| | | 43 | na | na | na | na | na | na | Na |
| | BHD | 72 | na | na | na | na | na | na | Na |
| | | 43 | na | na | na | na | na | na | Na |
| | Height | 72 | na | na | na | na | na | na | Na |
| | | 43 | na | na | na | na | na | na | Na |
| | BHD & RHD | 72 | 0.98 | 1.2 | 0.4 | 0.3 | 1.8 | 0.99 | Very good |
| | | 43 | 0.99 | 1.6 | 0.5 | 0.4 | 1.8 | 1.00 | Satisfactory |
| | Height & BHD | 72 | 0.99 | 0.7 | 0.3 | 0.3 | 2.1 | 0.99 | Acceptable |
| | | 43 | 0.99 | 0.7 | 0.3 | 0.3 | 1.2 | 0.99 | Satisfactory |
| | Height & RHD | 72 | 1.00 | 0.1 | 0.1 | 0.1 | 0.2 | 1.00 | Very good |
| | | 43 | 1.00 | 0.1 | 0.1 | 0.1 | 1.0 | 1.00 | Very good |
| Power: two terms | RHD | 72 | 0.98 | 2.5 | 0.6 | 0.6 | 1.3 | 0.99 | Satisfactory |
| | | 43 | 0.97 | 2.6 | 0.6 | 0.6 | 2.2 | 0.98 | Very good |
| | BHD | 72 | 0.96 | 2.4 | 0.6 | 0.5 | 0.0 | 0.98 | Acceptable |
| | | 43 | 0.96 | 2.6 | 0.6 | 0.6 | 2.7 | 0.96 | Satisfactory |
| | Height | 72 | 0.97 | 12,883 | 43 | 38 | 1.0 | 0.99 | Very good |
| | | 43 | 0.97 | 15,558 | 47 | 41 | 3.1 | 0.98 | Very good |
| | BHD & RHD | 72 | 1.00 | 0.2 | 0.1 | 0.1 | 1.4 | 1.00 | Very good |
| | | 43 | 1.00 | 0.2 | 0.2 | 0.1 | 1.0 | 1.00 | Very good |
| | Height & BHD | 72 | 1.00 | 0.2 | 0.2 | 0.1 | 1.5 | 0.99 | Satisfactory |
| | | 43 | 1.00 | 0.4 | 0.2 | 0.1 | 2.5 | 0.98 | Acceptable |
| | Height & RHD | 72 | 1.00 | 0.1 | 0.1 | 0.1 | 0.2 | 1.00 | Very good |
| | | 43 | 1.00 | 0.1 | 0.1 | 0.1 | 1.1 | 1.00 | Very good |
| Rational | RHD | 72 | 1.00 | 0.8 | 0.3 | 0.3 | 0.3 | 0.99 | Satisfactory |
| | | 43 | 1.00 | 1.5 | 0.5 | 0.3 | 2.3 | 0.99 | Acceptable |
| | BHD | 72 | 0.99 | 0.8 | 0.3 | 0.3 | 1.5 | 0.98 | Poor |
| | | 43 | 1.00 | 1.7 | 0.5 | 0.3 | 4.0 | 0.98 | Poor |
| | Height | 72 | 0.00 | 479,485 | 262 | 231 | 3.1 | 0.00 | Poor |
| | | 43 | 0.00 | 480,081 | 262 | 236 | 3.2 | 0.00 | Poor |
| | BHD & RHD | 72 | 1.00 | 0.2 | 0.2 | 0.1 | 1.2 | 1.00 | Very good |
| | | 43 | 1.00 | 0.1 | 0.1 | 0.1 | 0.0 | 1.00 | Very good |
| | Height & BHD | 72 | 1.00 | 0.2 | 0.2 | 0.1 | 1.5 | 0.99 | Satisfactory |
| | | 43 | 1.00 | 0.3 | 0.2 | 0.1 | 2.4 | 0.99 | Acceptable |
| | Height & RHD | 72 | 1.00 | 0.1 | 0.1 | 0.1 | 0.2 | 1.00 | Very good |
| | | 43 | 1.00 | 0.1 | 0.1 | 0.1 | 1.2 | 1.00 | Very good |
| Sum of Sine | RHD | 72 | 1.00 | 0.4 | 0.2 | 0.2 | 0.3 | 0.99 | Very good |
| | | 43 | 1.00 | 0.6 | 0.3 | 0.2 | 2.1 | 0.99 | Satisfactory |
| | BHD | 72 | 1.00 | 0.3 | 0.2 | 0.2 | 1.5 | 0.98 | Acceptable |
| | | 43 | 1.00 | 0.6 | 0.3 | 0.2 | 3.5 | 0.98 | Acceptable |
| | Height | 72 | 1.00 | 921 | 11 | 11 | 0.0 | 1.00 | Very good |
| | | 43 | 1.00 | 1421 | 14 | 9.0 | 0.9 | 1.00 | Very good |
| | BHD & RHD | 72 | 1.00 | 0.2 | 0.2 | 0.1 | 1.3 | 1.00 | Very good |
| | | 43 | 1.00 | 0.1 | 0.1 | 0.1 | 0.0 | 1.00 | Very good |
| | Height & BHD | 72 | 1.00 | 0.2 | 0.2 | 0.1 | 1.5 | 0.99 | Satisfactory |
| | | 43 | 1.00 | 0.3 | 0.2 | 0.1 | 2.2 | 0.99 | Satisfactory |
| | Height & RHD | 72 | 1.00 | 0.1 | 0.1 | 0.1 | 0.2 | 1.00 | Very good |
| | | 43 | 1.00 | 0.1 | 0.1 | 0.1 | 1.2 | 1.00 | Very good |
| Linear Fit | RHD | 72 | 0.98 | 1.0 | 0.4 | 0.3 | 1.0 | 0.99 | Very good |
| | | 43 | 1.00 | 6.0 | 0.9 | 0.7 | 0.0 | 0.97 | Poor |
| | BHD | 72 | 0.98 | 1.2 | 0.4 | 0.4 | 0.6 | 0.99 | Very good |
| | | 43 | 1.00 | 7.1 | 1.0 | 0.8 | 3.5 | 0.98 | Poor |
| | Height | 72 | 0.99 | 6198 | 30 | 27 | 0.6 | 0.99 | Satisfactory |
| | | 43 | 1.00 | 33,105 | 69 | 51 | 1.1 | 0.97 | Poor |
| | BHD & RHD | 72 | 0.95 | 6.4 | 1.0 | 0.8 | 3.9 | 0.97 | Poor |
| | | 43 | 1.00 | 6552.4 | 30.6 | 21.3 | 45.1 | −0.03 | Poor |
| | Height & BHD | 72 | 0.97 | 1.8 | 0.5 | 0.4 | 2.0 | 0.98 | Acceptable |
| | | 43 | 1.00 | 65.9 | 3.1 | 2.2 | 17.6 | 0.55 | Poor |
| | Height & RHD | 72 | 0.98 | 1.7 | 0.5 | 0.4 | 0.4 | 0.99 | Satisfactory |
| | | 43 | 1.00 | 60.3 | 2.9 | 2.1 | 11.6 | 0.68 | Poor |
| Polynomial: first degree | RHD | 72 | 0.98 | 2.6 | 0.6 | 0.6 | 1.3 | 0.99 | Satisfactory |
| | | 43 | 0.97 | 2.7 | 0.6 | 0.6 | 2.2 | 0.98 | Very good |
| | BHD | 72 | 0.96 | 2.4 | 0.6 | 0.5 | 0.0 | 0.98 | Acceptable |
| | | 43 | 0.96 | 2.7 | 0.6 | 0.6 | 2.7 | 0.96 | Satisfactory |
| | Height | 72 | 0.97 | 13,187 | 43 | 39 | 1.0 | 0.99 | Very good |
| | | 43 | 0.97 | 15,939 | 48 | 42 | 3.2 | 0.98 | Very good |
| | BHD & RHD | 72 | 1.00 | 0.2 | 0.2 | 0.1 | 1.3 | 1.00 | Very good |
| | | 43 | 1.00 | 0.1 | 0.1 | 0.1 | 0.0 | 1.00 | Very good |
| | Height & BHD | 72 | 1.00 | 0.2 | 0.2 | 0.1 | 1.3 | 0.99 | Satisfactory |
| | | 43 | 1.00 | 0.2 | 0.2 | 0.1 | 1.4 | 0.99 | Acceptable |
| | Height & RHD | 72 | 1.00 | 0.1 | 0.1 | 0.1 | 0.3 | 1.00 | Very good |
| | | 43 | 1.00 | 0.1 | 0.1 | 0.1 | 1.1 | 1.00 | Very good |

**Table A2.** *Cont.*

| Model | Variable | Data Gap Representation | $R^2$ | SSE | RMSE | MAE | SB [%] | CCC | Label |
|---|---|---|---|---|---|---|---|---|---|
| Polynomial: second degree | RHD | 72 | 1.00 | 0.6 | 0.3 | 0.2 | 0.3 | 0.99 | Satisfactory |
| | | 43 | 1.00 | 0.9 | 0.4 | 0.2 | 2.1 | 0.99 | Satisfactory |
| | BHD | 72 | 1.00 | 0.5 | 0.3 | 0.2 | 1.5 | 0.98 | Acceptable |
| | | 43 | 1.00 | 0.9 | 0.4 | 0.2 | 3.6 | 0.98 | Poor |
| | Height | 72 | 1.00 | 1774 | 16 | 15 | 0.0 | 1.00 | Very good |
| | | 43 | 1.00 | 2636 | 19 | 13 | 0.9 | 1.00 | Very good |
| | BHD & RHD | 72 | 1.00 | 0.1 | 0.1 | 0.1 | 1.4 | 1.00 | Very good |
| | | 43 | 1.00 | 0.1 | 0.1 | 0.1 | 0.8 | 1.00 | Very good |
| | Height & BHD | 72 | 1.00 | 0.2 | 0.2 | 0.1 | 1.5 | 0.99 | Satisfactory |
| | | 43 | 1.00 | 0.3 | 0.2 | 0.1 | 2.3 | 0.99 | Acceptable |
| | Height & RHD | 72 | 1.00 | 0.1 | 0.1 | 0.1 | 0.2 | 1.00 | Very good |
| | | 43 | 1.00 | 0.1 | 0.1 | 0.1 | 1.1 | 1.00 | Very good |

\* A negative R-square is possible if the model does not contain a constant term and the fit is poor (worse than just fitting the mean); na: not available; RHD: root height diameter; BHD: breast height diameter.

**Table A3.** The goodness of validation of all applied interpolation models in terms of the coefficient of determination ($R^2$), sum of squared errors (SSE), root-mean-square error (RMSE), mean absolute error (MAE), as well as the concordance correlation coefficient (CCC) and simulation bias (SB) from the observations.

| Model | Variable | Data Gap Representation | $R^2$ | SSE | RMSE | MAE | SB [%] | CCC | Label |
|---|---|---|---|---|---|---|---|---|---|
| Interpolant: Nearest Neighbor | RHD | 72 | 1.00 | 4.8 | 0.8 | 0.5 | 2.3 | 0.97 | Poor |
| | | 43 | 1.00 | 12.6 | 1.3 | 0.9 | 0.4 | 0.93 | Poor |
| | BHD | 72 | 1.00 | 4.0 | 0.8 | 0.4 | 1.0 | 0.97 | Poor |
| | | 43 | 1.00 | 8.9 | 1.1 | 0.8 | 0.0 | 0.92 | Poor |
| | Height | 72 | 1.00 | 32,985 | 69 | 36 | 1.5 | 0.97 | Poor |
| | | 43 | 1.00 | 71,174 | 101 | 71 | 1.4 | 0.93 | Poor |
| | BHD & RHD | 72 | 1.00 | 6.0 | 0.9 | 0.5 | 6.8 | 0.97 | Poor |
| | | 43 | 1.00 | 12.6 | 1.3 | 0.9 | 0.4 | 0.93 | Poor |
| | Height & BHD | 72 | 1.00 | 6.3 | 0.9 | 0.5 | 8.7 | 0.95 | Poor |
| | | 43 | 1.00 | 8.9 | 1.1 | 0.8 | 0.0 | 0.92 | Poor |
| | Height & RHD | 72 | 1.00 | 6.5 | 1.0 | 0.5 | 5.2 | 0.97 | Poor |
| | | 43 | 1.00 | 12.6 | 1.3 | 0.9 | 0.4 | 0.93 | Poor |
| Interpolant: Linear | RHD | 72 | 1.00 | 0.3 | 0.2 | 0.1 | 0.8 | 1.00 | Very good |
| | | 43 | 1.00 | 0.5 | 0.3 | 0.2 | 0.4 | 0.99 | Very good |
| | BHD | 72 | 1.00 | 0.1 | 0.1 | 0.1 | 0.9 | 0.99 | Satisfactory |
| | | 43 | 1.00 | 0.4 | 0.2 | 0.2 | 0.0 | 0.98 | Satisfactory |
| | Height | 72 | 1.00 | 211 | 5.0 | 2.0 | 0.4 | 1.00 | Very good |
| | | 43 | 1.00 | 3154 | 21 | 13 | 1.4 | 1.00 | Very good |
| | BHD & RHD | 72 | 1.00 | 0.3 | 0.2 | 0.1 | 1.5 | 1.00 | Very good |
| | | 43 | 1.00 | 0.0 | 0.1 | 0.0 | 0.2 | 1.00 | Very good |
| | Height & BHD | 72 | 1.00 | 0.1 | 0.1 | 0.1 | 1.3 | 0.99 | Satisfactory |
| | | 43 | 1.00 | 0.2 | 0.2 | 0.1 | 1.6 | 0.99 | Satisfactory |
| | Height & RHD | 72 | 1.00 | 0.2 | 0.2 | 0.1 | 0.5 | 1.00 | Very good |
| | | 43 | 1.00 | 0.1 | 0.1 | 0.1 | 1.1 | 1.00 | Very good |
| Interpolant: Cubic | RHD | 72 | 1.00 | 0.3 | 0.2 | 0.1 | 1.2 | 1.00 | Very good |
| | | 43 | 1.00 | 0.9 | 0.4 | 0.2 | 2.1 | 0.99 | Satisfactory |
| | BHD | 72 | 1.00 | 0.1 | 0.1 | 0.1 | 0.7 | 0.99 | Very good |
| | | 43 | 1.00 | 0.9 | 0.4 | 0.2 | 3.6 | 0.98 | Poor |
| | Height | 72 | 1.00 | 594 | 9.0 | 4.0 | 0.3 | 1.00 | Very good |
| | | 43 | 1.00 | 2636 | 19 | 13 | 0.9 | 1.00 | Very good |
| | BHD & RHD | 72 | 1.00 | 0.7 | 0.3 | 0.2 | 2.6 | 1.00 | Satisfactory |
| | | 43 | 1.00 | 0.1 | 0.1 | 0.1 | 0.8 | 1.00 | Very good |
| | Height & BHD | 72 | 1.00 | 0.1 | 0.1 | 0.1 | 1.0 | 0.99 | Very good |
| | | 43 | 1.00 | 0.3 | 0.2 | 0.1 | 2.3 | 0.99 | Acceptable |
| | Height & RHD | 72 | 1.00 | 0.4 | 0.2 | 0.1 | 1.8 | 1.00 | Very good |
| | | 43 | 1.00 | 0.1 | 0.1 | 0.1 | 1.1 | 1.00 | Very good |
| Interpolant: PCHIP | RHD | 72 | 1.00 | 0.2 | 0.2 | 0.1 | 0.8 | 1.00 | Very good |
| | | 43 | 1.00 | 1.0 | 0.4 | 0.2 | 2.1 | 0.99 | Satisfactory |
| | BHD | 72 | 1.00 | 0.1 | 0.1 | 0.1 | 1.2 | 0.99 | Satisfactory |
| | | 43 | 1.00 | 1.1 | 0.4 | 0.2 | 3.6 | 0.98 | Poor |
| | Height | 72 | 1.00 | 60 | 3.0 | 2.0 | 0.0 | 1.00 | Very good |
| | | 43 | 1.00 | 3398 | 22 | 16 | 0.9 | 1.00 | Very good |
| | BHD & RHD | 72 | 1.00 | 0.4 | 0.2 | 0.1 | 1.8 | 1.00 | Very good |
| | | 43 | 1.00 | 0.1 | 0.1 | 0.1 | 0.8 | 1.00 | Very good |
| | Height & BHD | 72 | 1.00 | 0.1 | 0.1 | 0.1 | 1.3 | 0.99 | Satisfactory |
| | | 43 | 1.00 | 0.3 | 0.2 | 0.1 | 2.3 | 0.99 | Acceptable |
| | Height & RHD | 72 | 1.00 | 0.2 | 0.2 | 0.1 | 0.8 | 1.00 | Very good |
| | | 43 | 1.00 | 0.1 | 0.1 | 0.1 | 1.1 | 1.00 | Very good |

RHD: root height diameter; BHD: breast height diameter.

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
