# Peer review of "Handling Data Gaps in Reported Field Measurements of Short Rotation Forestry"

_data, 2019_

Round 1

Reviewer 1 Report

In this research, authors used only averaged value of all trees in a year (hence, single value for an attribute in a given year) for given year. For example, there is a single value for tree height in year 2008. Similarly other other attributes (root height diameter, breast height diameter ) have a single value for each year. Since there are only 3 values when only 43 % and 5 values when 72% of the original data set were used. I would strong urged authors to use all observed values (each attribute value for each tree in study area for each year). For example, if there were 100 trees, then use 100 height values in each year. In case of 42% of original data, authors would have 100*3 (100 trees and 3 years) = 300 tree height data. Similarly, other attributes. General comments: 1. Please report study area size (how big was study area)? 2. Total number of trees (if possible) in the study area? 3. In Figure 1: Use different color or symbol for missing data and existing data. After plotting line, its really hard to see if there is any data existing data

Author Response

Thank you so much for taking the time to review our paper "Handling Data Gaps in Reported Field Measurements of Short Rotation Forestry" and for the kind feedback. 

According to your suggestions, we have added more of the available information about the study area and variability of measurements in the investigated data sets. Additionally, the figures now use different symbols and colour for the missing and existing data.

Moreover, we have examined your Grading from the Open Review and attempted to correct accordingly. Therefore, the Results, Discussion, and Conclusions sections were enhanced, which will hopefully make these sections easier to understand for the reader. 

I hope our corrections are to your satisfaction and we look forward to hearing from you. Thank you again for your feedback.

With best regards and on behalf of the co-authors,
Diana Seserman

Reviewer 2 Report

This paper describes a comparison between several gap filling methods for missing data in forest study. More data will be required for statistical soundness.

Major concerns

In my humble opinion, seven points are not sufficient for statistical meaning. More data should be used for performance test of the gap filling methods.

Minor concerns

MAE stands for mean absolute error, not mean average error.

Author Response

Thank you so much for taking the time to review our paper "Handling Data Gaps in Reported Field Measurements of Short Rotation Forestry" and for the kind feedback. 

According to your suggestions, we have added more of the available information about the study area and variability of measurements in the investigated data sets. Minor concerns were also addressed.

Moreover, we have examined your Grading from the Open Review and attempted to correct accordingly. Therefore, the Results, Discussion, and Conclusions sections were enhanced, which will hopefully make these sections easier to understand for the reader. 

I hope our corrections are to your satisfaction and we look forward to hearing from you. Thank you again for your feedback.

With best regards and on behalf of the co-authors,
Diana Seserman